# Digital Health Interventions for Adolescents with Long-Term Health Conditions in South Africa: A Scoping Review

**DOI:** 10.3390/ijerph22010002

**Published:** 2024-12-24

**Authors:** Talitha Crowley, Lwandile Tokwe, Leonie Weyers, Rukshana Francis, Charné Petinger

**Affiliations:** 1School of Nursing, University of the Western Cape, Bellville 7535, South Africa; 3749018@myuwc.ac.za (L.W.); ruadams@uwc.ac.za (R.F.); 2HIV Mental Health Research Unit, University of Cape Town, Cape Town 7700, South Africa; lwadz.tokwe@uct.ac.za; 3School of Public Health, University of the Western Cape, Bellville 7535, South Africa; 3720520@myuwc.ac.za

**Keywords:** adolescents, chronic disease, digital health, long-term health conditions, scoping review, South Africa

## Abstract

Adolescents with long-term health conditions may benefit from digital health interventions (DHIs) to support self-management. The study aimed to map the current research on DHIs for adolescents with long-term conditions in South Africa, focusing on the types of interventions, targeted chronic conditions, and reported outcomes. The scoping review was conducted following the Joanna Briggs Institute (JBI) guidelines and the Preferred Reporting Items for Systematic Reviews and Meta-Analyses Extension for Scoping Reviews (PRISMA-ScR) checklist. Searches were conducted in electronic databases such as EBSCOHost (CINAHL, MEDLINE, Academic Search Ultimate, and APA PSycArticles), Wiley Online Library, and PubMed for articles published between 2014 and 2024. Studies that (1) involved adolescents with a long-term health condition (aged 15–24) residing in South Africa, (2) reported on the use of digital health technology, and (3) provided empirical evidence were included. Nine studies were included in the analysis, focusing primarily on HIV, depression/anxiety, and diabetes. Most interventions utilized WhatsApp, SMS, or social media to provide peer or healthcare worker support. Process outcomes like acceptability and feasibility dominated, with limited data on effectiveness. DHIs show potential for supporting adolescent health but cover a limited number of long-term health conditions and face barriers to effective implementation. Affordable, context-specific solutions co-designed with adolescents are crucial to enhance engagement and ensure scalability in the South African context. Registration: The protocol was registered on Open Science Framework.

## 1. Introduction

There is a growing, global increase in adolescents and children living with a chronic or long-term health condition [1]. A long-term health condition affects an adolescent for substantial periods, ranging from more than a year to lifelong, potentially impacting their physical or psychological development and requiring continuous and comprehensive health care [2,3]. We use the term “long-term health conditions” to refer to both chronic, lifelong conditions and those that may not be lifelong. Long-term health conditions can result in early mortality and disproportionately affect low-resource communities [4].

In 2022, adolescents aged 10 to 19 years made up 10,468 (17.4%) of the population of 59,894 in South Africa [5,6]. Adolescence is typically divided into three stages: early adolescence (ages 10–14), middle adolescence (ages 15–17), and late adolescence/young adulthood (ages 18–24) [7]. It is a critical developmental period in which many behaviors are established, providing an opportunity to promote the adoption of healthy habits [8]. Consequently, adolescents living with chronic or long-term health conditions face difficulties and developmental challenges in the social, biological, and psychological spheres [9].

South Africa has a convergence of non-communicable and communicable diseases [10,11]. About 25% of adolescents in South Africa live with a long-term health condition [2]. Long-term health conditions encompass chronic disease (communicable or non-communicable diseases, e.g., asthma, chronic lung disease, human immunodeficiency virus (HIV), tuberculosis (TB), and diabetes) and other conditions such as physical disabilities, mental health conditions, and developmental disorders. In South Africa, HIV and non-communicable diseases (NCDs) account for 38% of deaths among adolescent girls aged 15 to 19 years and 29% of deaths among adolescent boys in the same age group [5]. In a study conducted in Cape Town, the medical records of 491 adolescents living with HIV (ALHIV) aged 10 to 24 were reviewed. The findings revealed that 11% had a documented NCD, with chronic respiratory diseases (60%) and mental disorders (37%) being the most prevalent [8].

Recent advances in digital health technology have led to the proliferation of interventions. The health outcomes of adolescents living with long-term health conditions are generally poor compared with children and adults [12]. Digital health holds promise to address the unique needs of these adolescents to promote disease self-management and overall health outcomes, particularly as adolescents are known to be persistent users of digital platforms [13,14,15]. Furthermore, technological or digital platforms for self-management may be an effective strategy for adolescents and youth, particularly when integrated with support from family members, peers, and healthcare workers [16]. A systematic review to evaluate the utility and effectiveness of mobile and web-based health applications that support self-management and transition in adolescents and youth with chronic physical health illnesses reported limited data on effectiveness but attributed this to design and evaluation limitations [17]. Other evidence has shown that digital technologies and interventions can improve the health outcomes of people living with chronic conditions in low- and middle-income countries and may provide a cheaper and easier addition to healthcare in these highly burdened healthcare systems [18].

Adolescents with long-term health conditions in South Africa face various socio-economic and health care access disparities [12]. It is reported that adolescents in South Africa feel left behind and alienated from digital advances such as artificial intelligence [19]. Further, little is known about the current digital landscape of digital health interventions (DHIs) for adolescents with long-term health conditions [15]. A scoping review on digital health for self-management in sub-Saharan Africa (SSA) found that although it is an emerging alternative approach to chronic disease management, due to the unique characteristics of digital health users in SSA, technologies and content should be tailored for this region [20]. This scoping review was not specific to the adolescent population and focused only on non-communicable diseases.

A preliminary search on MEDLINE (EBSCOhost), PubMed, the Cochrane Database of Systematic Reviews, and JBI Evidence Synthesis revealed no current or underway systematic reviews or scoping reviews on DHIs for adolescents with chronic or long-term health conditions in South Africa.

The objective of the review was to identify and map the current research on digital health interventions for adolescents with long-term health conditions in South Africa, focusing on types of interventions, targeted chronic conditions, and reported outcomes.

The review questions were as follows:What types of digital health interventions have been studied for adolescents with long-term health conditions in South Africa?Which long-term health conditions are targeted by these digital health interventions?What are the reported outcomes of these interventions?

## 2. Materials and Methods

### 2.1. Design

The review followed the Joanna Briggs Institute (JBI) guidelines [21] and the Preferred Reporting Items for Systematic Reviews and Meta-Analyses Extension for Scoping Reviews (PRISMA-SCR) guidelines for scoping reviews [22]. It was registered on the Open Science Framework (https://doi.org/10.17605/OSF.IO/BZQ6Y) (accessed on 5 July 2024).

### 2.2. Inclusion Criteria

Types of participants—The participants were adolescents with long-term health conditions aged 15 to 24 years. This review focused on the age group 15–24 years as individuals in this range are likely to benefit from DHIs during their transition into adulthood. We defined a long-term health condition as a condition acquired at any time from birth to adolescence that requires long-term (>12 months) self-management strategies such as behavior change and treatment adherence [2]. It can include but is not limited to the following: allergic (asthma or dermatitis), infectious (HIV and TB (if diagnosed >12 months)), auto-immune (diabetes and juvenile arthritis), psychiatric (depression or other mental health diagnoses), nutritional (e.g., obesity), cardiovascular (hypertension or heart disease), and neurological (epilepsy) conditions.

Concept, comparison and outcomes—The review included any intervention or study delivering health-focused digital technology for adolescents living with a long-term health condition. This includes but is not limited to mobile health apps, telemedicine, online support groups, and digital monitoring tools. Digital technology encompasses various technologies such as mobile health applications, online platforms, text messaging, interactive voice response systems, and other innovative tools [23].

We included all studies with or without a comparison. The outcomes included experiences, acceptability, usability, and any health-related outcomes.

Context—Studies conducted in South Africa in all types of healthcare settings or in community settings were included.

Timeframe—Studies were conducted between 2014 and 2024. By focusing on the last decade, the review aimed to map the most recent evidence of implementation and adoption in the South African context.

Types of evidence sources—All types of empirical studies were included, such as randomized controlled trials, cohort studies, case studies, and qualitative studies.

Language—Studies published in the English language were included as English is the lingua franca and primary language for academic publishing in South Africa.

### 2.3. Exclusion Criteria

We excluded studies published earlier than 2014, not in the English language, and outside of South Africa. The search was limited to the past 10 years to capture the most recent evidence, focusing on South Africa to ensure relevance to the local context.

### 2.4. Search Strategy

The search strategy was developed through the use of PubMed to determine search strings and thesaurus terms identified through Academic Search Ultimate (Box 1 and Box 2). Limiters included the timeframe (2014–2024), language (English), and age range (15–24 years). The following databases were used: EBSCOHost (CINAHL, MEDLINE, Academic Search Ultimate, APA PSycArticles), Wiley Online Library, and PubMed, providing coverage of the health, technology-related, and psychosocial literature and aligning with the JBI guidelines for multidisciplinary and topic-specific databases [21]. Additional articles were identified through Google Scholar and reference mining to ensure that all eligible studies were included in our review. The initial search strategy was quality assured by an information specialist.

Box 1PubMed search string.(((((((((((((((digital technology) OR (mHealth)) OR (digital health)) OR (eHealth)) OR (online)) OR (social media)) OR (mobile phone)) OR (mobile applications) OR (smartphone)) AND (Youth)) OR (adolescent)) OR (young adult)) AND (chronic disease) OR (chronically ill) OR (disease)) OR (long-term illness) AND (South Africa)))) AND ((english[Filter]) AND (adolescent[Filter] OR youngadult[Filter]) AND (2014:2024[pdat])) AND ((english[Filter]) AND (adolescent[Filter] OR youngadult[Filter])) AND ((english[Filter]) AND (adolescent[Filter] OR youngadult[Filter]))

Box 2EBSCOHost, Wiley Online Library, and Google Scholar Search string.(((((((((((((((digital technology) OR (mHealth)) OR (digital health)) OR (eHealth)) OR (online)) OR (social media)) OR (mobile phone)) OR (mobile applications) OR (smartphone)) AND (Youth)) OR (adolescent)) OR (young adult)) AND (chronic disease) OR (chronically ill) OR (disease)) OR (long-term illness) AND (South Africa))))youngadult[Filter]))

### 2.5. Study Selection

After the search, all identified articles were uploaded into [24], a software platform. Covidence facilitated the removal of duplicate articles and guided the screening process, from the initial screening to data extraction. Two reviewers (L.T. and L.W.) independently screened titles and abstracts for inclusion. Full texts of potentially relevant studies were retrieved and assessed for eligibility by two reviewers (R.F. and C.P.). Discrepancies were resolved through discussion with another reviewer (T.C.). The search results are presented in a PRISMA diagram with the reasons for exclusion indicated.

### 2.6. Data Charting

A data extraction form was used to chart information on study characteristics (authors, year of publication, study design, and aim of study), population (age and specific long-term health condition), interventions (e.g., type of digital technology and duration), outcomes, and barriers/facilitators. Two reviewers (R.F. and C.P.) independently extracted data. Discrepancies were resolved between the two reviewers, and where necessary, a third reviewer (T.C.) was consulted.

### 2.7. Data Synthesis

A descriptive synthesis was conducted to summarize the results. We used an adapted logic model for digital health interventions previously developed by the authors to map the following: (1) the challenge necessitating the DHI; (2) the components of the DHI; (3) the process, proximal, and distal outcomes; and (4) the contextual factors influencing the intervention outcomes (Figure 1) [25]. The findings are presented in tables, figures, and narrative form as per guidelines [26].

### 2.8. Ethics and Dissemination

This review was part of a larger study investigating self-management support for adolescents with long-term health conditions in South Africa and was approved by the Biomedical Health Research Ethics Committee of the University of the Western Cape (BM24/5/9).

## 3. Results

### 3.1. Selection of Sources of Evidence

Following the database search on 29 July 2024, 3570 studies were screened, after duplicate studies were removed by Covidence, of which 20 were included for full-text review. After applying eligibility criteria, nine studies were included in the review, of which one was ongoing (Figure 2). Of the database searches, 3251 results were from PubMed, 283 were yielded from EBSCOHost, 26 were from Wiley Online Library, and 14 results were from Google Scholar. 

### 3.2. Characteristics of Included Studies

Of the nine studies included, most studies were conducted in the Western Cape Province [27,28,29,30]. Three studies [31,32,33] were randomized controlled trials (RCTs), while the rest were observational quantitative, qualitative, or mixed-methods studies. One study [32] was ongoing (Table 1).

Six studies addressed HIV [27,28,29,30,31,34], two studies focused on depression and anxiety [32,33], and one study targeted diabetes [35]. The targeted adolescent age range was 12 to 25, which included middle and late adolescence and early adulthood. The study duration ranged from one month to 17 months, and the number of participants varied from 7 to 371.

Six DHIs addressed challenges related to mental health and/or isolation [27,28,29,31,32,33]. Four studies addressed challenges of adherence and/or retention in care [29,30,31,34], and three studies [28,30,31] focused on treatment outcomes as a challenge. Four studies addressed self-management or transition readiness [29,31,34,35].

**Table 1 ijerph-22-00002-t001:** Study characteristics.

Author (Year), Geographical Location	Study Design	Name of Intervention	Type of LTHC or Study Population and Sample Size	Duration in Months	Targeted Health System/Adolescent Challenge Addressed by Intervention
Zanoni et al. (2024), KwaZulu-Natal [31]	RCT	Interactive Transition Support for Adolescents with HIV (InTSHA)	Adolescents living with perinatally acquired HIV (APHIV) aged 15–19 years (N = 80)	17	Adherence and retention in care Treatment outcomes Mental health and isolation Self-management–transition support
Henwood et al. (2016), Western Cape [27]	Mixed methods	Khaya HIV Positive–Virtual Support Group	ALHIV aged 12–25 years (N = 60—quantitative; N = 12—qualitative); females, 63%, and males, 37%	13	Mental health and isolation–psychosocial support between club meetings
Atujuna et al. (2021), Gauteng and Western Cape. [28] Three of the selected clinics were located in an urban or peri-urban setting, although patients attending these clinics came from urban, semi-urban, and informal communities. One clinic was located in an urban informal township.	Pilot pretest-posttest	Project Khuluma–peer-to-peer mHealth intervention	ALHIV aged 15–25 years (N = 52); females, 56%, and males, 44%	6	Mental health and isolation–peer support Treatment outcomes
Williams et al. (2023), Gauteng [35]	Qualitative	Continuous Glucose Monitoring (CGM)	Adolescents living with type 1 diabetes using CGM aged 12–17 years (N = 7)	1	Self-management–glucose monitoring
Moffet et al. (2022), Mapumalanga [32], rural northeast of South Africa in the Bushbuckridge sub-district of Mpumalanga province	Pilot RCT	DoBat 3: digitally delivered behavioral activation therapy (BAT)	Adolescents aged 15–19 with mild to moderate depression (in progress)	6	Mental health and isolation–overcome depression and social/economic transition, socio-affective and cognitive processing Risky behavior
Mulawa et al. (2023), Western Cape [29]	Mixed methods beta testing	Masakhane Siphucule Impilo Yethu (MASI; Xhosa for “Let’s empower each other and improve our health”)	Adolescents living with perinatally acquired HIV (APHIV) aged 15–19 years (N = 14); female, 50%, and male, 50%	4	Mental health and isolation Adherence Self-management
Bergam et al. (2022), Kwazulu-Natal [34], urban township of KwaZulu-Natal, South Africa	Qualitative	InTSHA	Adolescents living with perinatally acquired HIV (APHIV) aged 15–19 years (N = 80); female, 61.9%, and male, 38.1%	17	Adherence and retention in care Self-management–HIV knowledge, behaviors, and attitudes
Bantjes et al. (2024), three universities in South Africa [33]	RCT	Two digital interventions: 1) remote digital gamified and group cognitive behavioral therapy skills training (CBTS) via the web and 2) SuperBetter gamified self-guided CBT skills training app	Undergraduate students with depression and/or anxiety aged 18–20 years (N = 371); female, 81.9%, and male, 18.1%	12	Mental health and isolation–anxiety and depression
Hacking et al. (2019), Western Cape [30], periurban informal settlement of Khayelitsha, Cape Town	Mixed methods	The Virtual Mentors Program	Newly diagnosed HIV positive aged 12–25 years; peer mentors and mentees (ALHIV) (N = 110); female, 95%, and male, 5%	14	Adherence and retention in care–linkage to care Treatment outcomes

### 3.3. Types and Components of Digital Health Interventions

All the digital health interventions except continuous glucose monitoring (CGM) [35] focused on providing healthcare worker or peer support, sharing experiences, or facilitating networking (Table 2). Four digital health interventions focused on information communication [31,32,33,34]. The two mental health interventions focused on the delivery of therapy [32,33].

Interventions were delivered on various digital health platforms including normal SMS, WhatsApp, and web-based and application-based platforms. Many of the interventions incorporated youth-focused platforms and technology designs such as audio-visuals to increase engagement. Five studies employed an interactive, group-based technology design [27,28,31,33,34], likely aimed at promoting peer and/or health care worker support. Three studies incorporated behavior or symptom tracking [29,33,35].

Five studies used SMS, WhatsApp, or social media [27,28,30,31,34]. The InTSHA (Interactive Transition Support for Adolescents with HIV [isiZulu for “youth”]) intervention used password-protected closed WhatsApp groups to share information including standardized text messages, images, and videos. Participants could also contact healthcare providers to ask health-related questions via separate private messages [31,34]. Henwood and colleagues [27] used password-protected chat rooms on a social media platform (MXit) moderated every weekday by a counsellor. Chats focused on health and social/family challenges. The Khuluma peer-to-peer intervention used group SMS messages to deliver group-based psychosocial support via mobile phones. Group SMS messaging was facilitated by a short code that allowed participants to send a message to a central number, from where it was distributed to their group. Groups of 8–15 participants were led on weekdays by trained peer mentors supported by professional counsellors over 12 weeks [28]. Hacking and colleagues [30] introduced a virtual mentor’s program using peer-to-peer mentoring for newly diagnosed youth. Mentors received message templates to interact with mentees via SMS with minimum required engagement.

Three interventions were app-based [29,32,33]. A study conducted by [32] developed a protocol for a pilot RCT to provide digital behavioral activation therapy (BAT) to adolescents with mild to severe depression. The intervention included six tailored modules of BAT via a smartphone application (the Kuamsha app) supported by trained peer mentors, implemented over 10 weeks. The Kuamsha app is an interactive narrative game that facilitates skills such as problem-solving, effective communication, getting enough sleep, and disengaging from rumination and uses game design elements to stimulate motivation and performance, including character personification, in-app points, and reminders/notifications. Similarly, reference [33] implemented an app-based gamified self-guided cognitive behavioral skills training (CBST) digital intervention (SuperBetter) in one of their study arms. Participants using the app were instructed to use it 10 min per day for 5 days per week over 10 weeks. A study by [33] compared 10-week web-based CBST skills training using trained facilitators with the SuperBetter app and MoodFlow mood monitoring control in university students with symptoms of depression and anxiety. Participants using the web program signed up for groups and were offered a one-on-one introduction followed by two sessions per week. Moodflow, which includes mood and sleep tracking and journaling was also used 10 min per day, 5 days per week, over 10 weeks.

A study by [29] developed a smartphone app for ALHIV (MASI) that promotes connection through a participant forum and provides resources on health, life skills, relationships, and well-being. Users interact anonymously and access customizable health tracking, peer discussions, and expert advice. The app offers activities like quizzes and goal setting on HIV management, relationships, and life skills, along with multimedia resources on similar topics.

One study by [35] used a digital continuous glucose monitoring (CGM) device. The study explored illness perceptions of adolescents living with type 1 diabetes using CGM. CGM provides a sustained visible measure of variations in glucose levels [35].

Only one study by [31] mentioned a participant-centered design in the development of the intervention, and one study [29] mentioned using a strength-based approach. None of the other interventions reported on the use of an underpinning theory.

### 3.4. Reported Outcomes

Of the nine studies included, eight studies reported on outcomes, and one study was ongoing (Table 3).

Seven studies reported on individual process outcomes such as engagement and acceptability or health systems level process outcomes, namely, feasibility [27,28,29,30,31,33,34]. Process outcomes such as engagement were generally high, although studies measured them differently. In the application-based intervention (MASI), 83% used the app every day, and in the peer-to-peer SMS intervention [28], 100% of participants sent at least one message, and 94% were retained in the intervention. Similarly, a study by [30] reported positive experiences from mentors and mentees in the Virtual Mentors program. The InTSHA WhatsApp-based intervention [31,34] reported that in at least half of the sessions, 60% of participants sent and responded to messages. A study by [27] utilized a chatroom service (MXit) as an addition to youth club meetings. While positive feedback was received and 57% of participants used the chatroom to obtain advice, a high number of participants ceased using the service. In addition, reference [31] also reported lower rates of feasibility (78%), compared with acceptability. This was due to participants experiencing power outages and poor internet connection and having to share cell phones [31]. A study by [29] reported positive experiences of using the MASI app, although some participants had network and technical challenges. Participants engaged in the various features, with most of the time spent on the health tracker (e.g., adherence tracking), followed by the forum. They accessed an average of one resource per day, and although few participants posted questions to the doctor (expert), most read the questions and responses [29]. Activities such as quizzes were some of the preferred features.

Four studies reported on immediate health promotion/disease prevention or self-management outcomes such as risk behaviors, treatment adherence, and retention in care (RiC) [30,31,34,35]. Although there were no significant differences in transition readiness, peer support, depression, self-esteem, drug or alcohol use, school attendance, connection to clinical staff, stigma, or RiC, the InTSHA WhatsApp-based intervention reported that RiC among adolescents attending three or more sessions (n = 30 [75%]) was 100%. Retention in care increased by 60% for each session attended [31]. Further, the qualitative evidence showed that it prompted discussions about sexual reproductive health (SHR) and changed attitudes, knowledge, and behavior [34]. A study by [30] reported increased linkage to care with the use of the Virtual Mentors program. A study conducted by [35] reported that adolescents with type 1 diabetes using CGM experienced an increased sense of control and that it assisted them to incorporate diabetes management into their daily routines.

Four studies reported on individual distal or mid-term health outcomes, such as viral suppression or mental health [30,31,33,35]. Although no statistically significant differences were found between the intervention and control groups, viral load suppression (VLS) in the InTSHA intervention was 83% (n = 25) for those who attended three or more sessions [31]. A study by [30] reported slightly higher VLS rates in the Virtual Mentors cohort. The study by [35] provided qualitative evidence that CGM in adolescents with type 1 diabetes might lead to improved health outcomes. In the study by [33], the Remote Group CBST and SuperBetter interventions showed significantly higher adjusted risk differences (ARDs) in combined anxiety and depression remission at both 3-month and 6-month follow-ups, compared with MoodFlow.

### 3.5. Contextual Factors Considered in the Design of Digital Health Interventions

Regarding the consideration of contextual factors in the design of the DHI, almost all the studies considered individual factors such as age and health literacy or technological ability, while few studies considered familial factors influencing technology use (Table 4). In the six studies considering age, the intervention was designed with the target audience being adolescents wherein their unique needs were addressed. For example, references [30,31] acknowledged that ALHIV benefit from peer support to improve their retention in care. Health literacy and technological ability were also considered in the design of the interventions; for example, reference [33] included a baseline assessment to ensure eligibility and workbooks to improve health literacy. Most studies also considered environmental factors such as safety, security, and access. A study by [35] considered geographical location, by only including participants who received their CGM and treatment at a specific facility. A study by [30] regarded adolescents’ preference for communicating via mobile devices and how they communicate (technology culture). Finally, reference [31] considered safety, security, and privacy by acknowledging that participants had to share their phones with others and thus used password-protected chats.

## 4. Discussion

Digital technologies of all different types have become a crucial resource, particularly in health settings such as primary health care (PHC), and their uptake is increasing, with the past 10 years seeing hasty integration of technology in different areas that support primary care and essential public health functions [36]. This scoping review mapped the current research on DHIs for adolescents with long-term health conditions in South Africa, focusing on types of interventions, targeted chronic conditions, and reported outcomes. We found only nine studies in South Africa that reported on DHIs for adolescents with long-term health conditions, addressing the second research question. The majority of the studies included in this review focused on ALHIV, followed by mental health conditions such as depression and anxiety and one study on diabetes. Most interventions identified mental health challenges and social isolation as key issues that need to be addressed by the DHI [27,28,29,31,32,33]. Other reviews also reported an increased prevalence of mental health among youth with chronic health conditions [37,38]. This emphasizes the need to focus on developing interventions that address mental health among adolescents with long-term health conditions.

The included studies indicated that most of the DHIs utilized existing platforms such as WhatsApp or social media [27,30,31,34]. This may be because these are widely used in the South African context [39,40]. There were only three studies that utilized applications [29,32,33]. These applications could integrate gamification features to improve engagement, but there is currently little evidence to show they lead to more engagement or better outcomes compared with SMS or WhatsApp-based interventions. None of the studies utilized artificial intelligence, e.g., chatbots. This may be because the technology is fairly new, and uptake has been slow in South Africa [41]. There is therefore a need to expand the scope of DHIs utilized in this context. A Kenyan study that developed and implemented a mobile application for sexual reproductive health education [42] also utilized social media, internet, and text messages for sharing health information with adolescents. In addition, a scoping review of studies from LMICs including Nigeria, Kenya, and China highlighted the use of digital health technology such as WhatsApp, text messages, and phone reminders to manage mental health [43]. However, despite these advancements in LMICs, mobile data affordability is a social determinant influencing access to and utilization of DHIs. A review study conducted in SSA [20] identified mobile data costs as a significant barrier preventing adolescents from effectively engaging with DHIs for managing long-term health conditions. In the South African context, the affordability of mobile data has been found to motivate adolescents to adopt DHIs [44]. These findings underscore the persistent challenges related to connectivity and affordability that hinder the uptake of DHIs for adolescents in SSA countries. There is a need to consider mobile data costs when designing these digital interventions for managing long-term health conditions for adolescents within SSA.

All the interventions aimed to incorporate peer or healthcare worker support within the DHI. The findings of this review show that DHIs are increasingly being integrated with peer and/or healthcare provider support. Subsequently, sharing of experiences and networking opportunities were provided through some of the interventions. This highlights the social aspects of health care not being removed through DHIs. Imperatively, a fear of this has been accounted for by previous research. In a systematic review conducted by [45] on social media use in health, it was found that social media may increase patients’ autonomy regarding their care, as well as perceived support. The additional psychosocial support provided to adolescents has been found to be beneficial, helping to reduce feelings of isolation and a lack of social connections by sharing experiences [46,47]. Further to psychological support, peer support has been shown to be beneficial for ALHIV, particularly in their retention in care [48,49]. This appears to suggest that DHIs have a positive impact on the psychological well-being and outcomes of adolescents. Therefore, these DHIs may be key to supporting adolescents living with long-term health conditions to develop self-management skills and transition to independent adult care [13].

In response to the third research question, the majority of the studies reported only on process outcomes such as acceptability and feasibility. This is expected since most studies were in the early development phases with few data on effectiveness. Digital health interventions were generally experienced as acceptable; however, implementation challenges such as network connectivity, device access, and privacy affected their feasibility. Adolescents require an increased focus on their privacy and safety on digital platforms [46]. Previous evidence found that adolescents fear that other people may have access to their information, which may result in unintended disclosure of their illness or personal information [48,49,50].

Two RCTs reported on effectiveness. One study [31] did not find any significant effect on immediate, mid-term, or distal outcomes, while another study [33] found that the CBST remote web-based programs and applications both improved symptoms of depression and anxiety. Observational and qualitative evidence indicated the potential benefit of DHIs for improving health behaviors and self-management. An integrative review of transition services and support for adolescents with long-term health conditions identified the need for effective transition and self-management support interventions to ensure positive outcomes [51]. We conclude that while DHIs hold promise for improving health outcomes for adolescents with long-term health conditions in South Africa, further rigorous research is needed.

Many of the studies considered contextual factors that influence the implementation of digital health interventions like geographic location, data/web accessibility, age, language, privacy/security, and health/digital literacy. Due to the diversity and disparity in South Africa, as well as the digital divide, more research is needed in various contexts on the use of digital interventions and how they can be used to support adolescents with long-term health conditions. Despite the evident acceptability of using DHIs, there remains a decreased uptake, particularly in the South African context [46]. Subsequently, while there is access to phones and the internet, the use and knowledge of DHIs remain low [46]. Given the reported challenges with implementation in the studies, it is important to further explore how contextual factors influence the implementation of digital health interventions across various contexts in South Africa. 

A strength of this study is that it is the first scoping review to map digital health interventions for adolescents with long-term health conditions in South Africa. Due to the focus on South Africa, the findings may not be generalizable to other LMICs. Further, while other contextually specific factors like gender and cultural and spiritual factors were not explicitly addressed in this scoping review, we acknowledge their significance in the South African context. This represents a gap that warrants further research. Additionally, due to the emerging nature of digital health interventions in South Africa and SSA, there are a limited number of local studies to compare with the findings of this review. We recommend that future research should include studies that implement and evaluate DHIs for adolescents with long-term health conditions. Additionally, there is a gap in developing DHIs that are tailored to South Africa’s diverse population and digital landscape. Designing interventions with adolescents in South Africa and stakeholders like community clinics could ensure that the digital tools meet their needs.

## 5. Conclusions

This review identified three key findings: First, most DHIs studied in South Africa utilized WhatsApp, SMS, or social media, while applications were less common, and advanced technologies like artificial intelligence were not used. Second, DHIs primarily targeted HIV, depression/anxiety, and diabetes to a lesser extent. Third, the reported outcomes predominantly focused on acceptability and feasibility, with limited evidence on effectiveness. While one study found no significant impact, another demonstrated improved mental health outcomes. To address these gaps, we recommend designing affordable, context-specific DHIs that integrate co-design processes with adolescents and stakeholders to enhance their relevance and uptake. Further research should prioritize rigorous evaluation of the effectiveness and scalability of DHIs across diverse health conditions and geographical contexts in South Africa.

## Figures and Tables

**Figure 1 ijerph-22-00002-f001:**
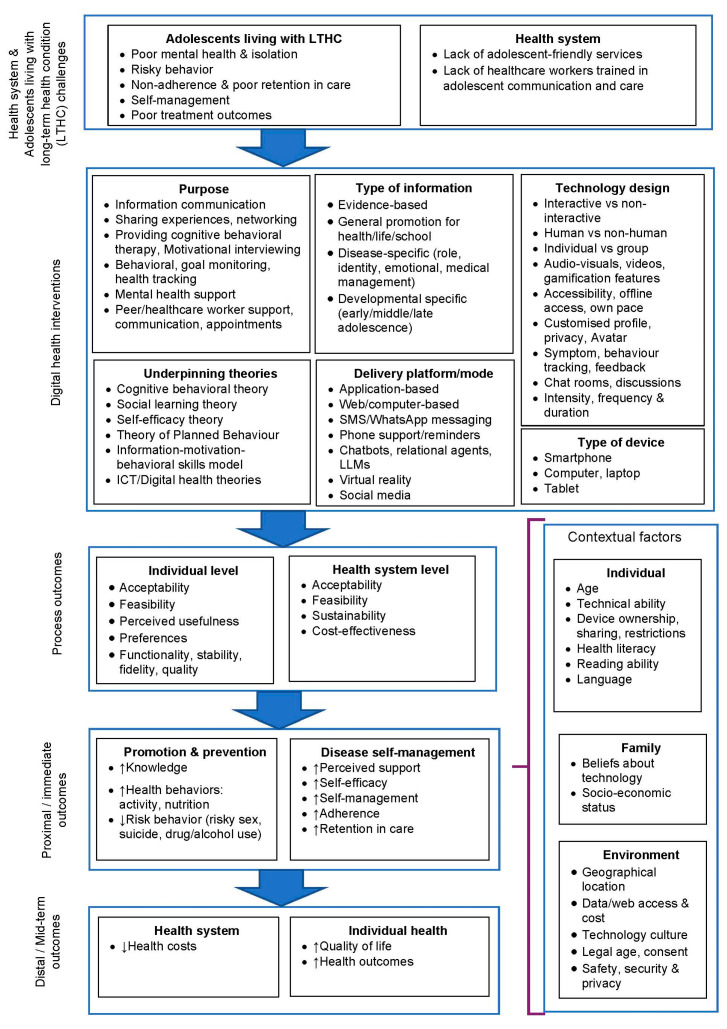
Logic model for digital health interventions for adolescents with long-term health conditions (adapted from Crowley et al., 2024) [25].

**Figure 2 ijerph-22-00002-f002:**
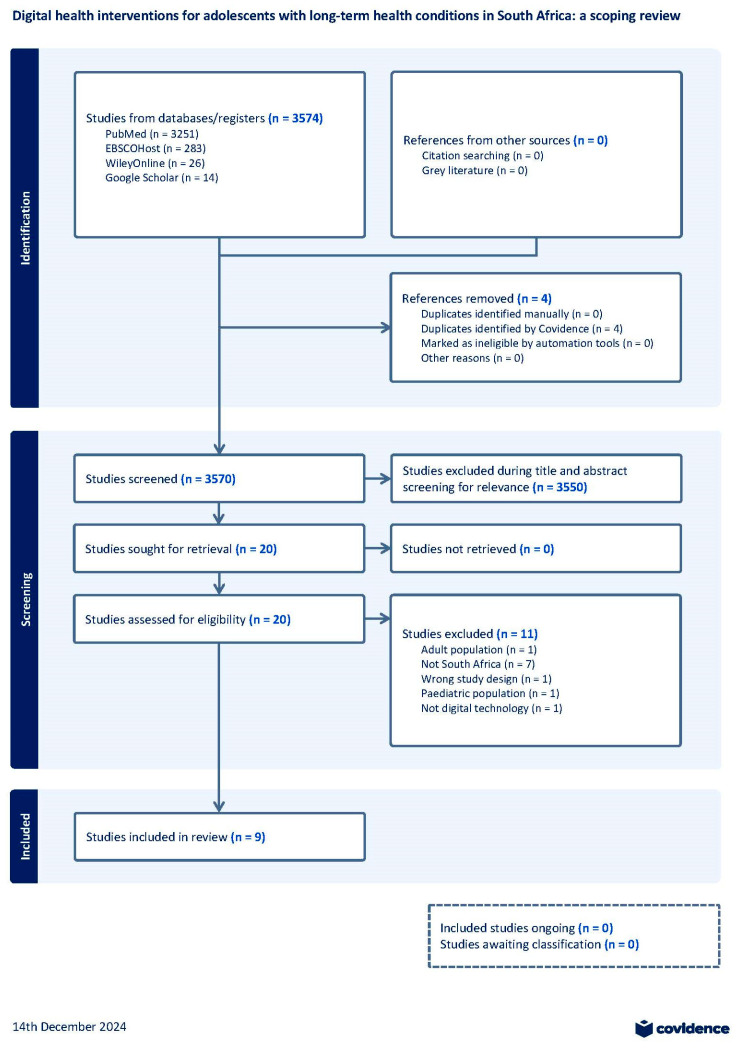
PRISMA diagram.

**Table 2 ijerph-22-00002-t002:** Types and components of digital health interventions.

Study	Purpose	Underpinning Theory	Delivery Platform/Mode	Technology Design	Type of Device
Zanoni et al., 2024 [31] and Bergham et al., 2022 [34]	Information communication–treatment adherence, stigma and self-efficacy, relationships, goals and future planning, and sexual and reproductive health (SRH) Sharing experiences/networking Peer/health care worker support–transition support	Participant- centered approach	WhatsApp messaging–long-term weekly group chats (12 sessions) facilitated by trained facilitators Separate group chat for caregivers	Interactive Groups and individual Standardized messaging Audio-visuals–videos and images	Smartphone
Henwood et al., 2016 [27]	Sharing experiences/networking Peer/health care worker support	None stated	Social media–MXit (SMS notifications when the counsellor joined the chats)	Interactive Groups	Smartphone
Atujuna et al., 2021 [28]	Peer support	None stated	SMS/messaging–facilitated Khuluma curriculum	Interactive Groups	Mobile phone
Williams et al., 2023 [35]	Behavioral/goal monitoring/health tracking	None stated	Device only	Symptom tracking/feedback–digital monitoring Individual	Glucose monitor
Moffett et al., 2022 [32]	Information communication Mental health support Providing cognitive behavioral therapy/motivational interviewing Peer/health care worker support	None stated	Application-based and phone calls	Interactive Individual Audio-visuals/videos/gamification features	Smartphone
Mulawa et al., 2023 [29]	Information communication–multimedia resources Sharing experiences/networking–asking experts	Strength-based approach	Application-based	Interactive activities Group and individual Behavior tracking–health tracker	Smartphone
Bantjes et al., 2024 [33]	Information communication Sharing experiences/networking Cognitive behavioral therapy/motivational interviewing Mental health support	None stated	One component web-based and the other application-based	Behavior tracking–Moodflow Interactive Groups and individual App–gamification	Web-based and smartphone
Hacking et al., 2019 [30]	Peer/health care provider support	None stated	SMS messaging	Interactive Individual	Mobile phone

**Table 3 ijerph-22-00002-t003:** Outcomes.

Study	Process Outcomes	Immediate Outcomes	Mid-Term/Distal Outcomes
Zanoni et al., 2024 [31]	***Individual*** Participation (sending and responding to messages) was scored as 60% in half of the sessions. ***Health systems*** 94% enrollment rate; 82% acceptability rate, although a 78% feasibility score	***Disease self-management*** No significant differences in peer support, depression, self-esteem, drug or alcohol use, school attendance, connection to clinical staff, stigma, or transition readiness No statistical difference in RiC among participants randomized to the InTSHA intervention compared with the control group (n = 39 [98%] vs. n = 35 [88%]; OR 5.6 [95%CI 0.6–50.0; *p =* 0.20])	***Individual health*** No statistical difference in viral suppression among adolescents randomized to the InTSHA intervention compared with the control [n = 29 [73%] vs. n = 33 [83%]; OR 0.6 (95% CI 0.2–1.6; *p =* 0.42)].
Bergam et al., 2022 [34]	***Individual*** Group facilitators play a novel and critical role in their sexual health education.	***Promotion and prevention*** Gaining a holistic understanding of SRH that changed attitudes, knowledge, and behaviors ***Disease self-management*** Allowed them to begin conversations about SRH with their caregivers, where discussion of SRH had previously been lacking empowerment	Not reported
Henwood et al., 2016 [27]	***Individual*** Moderate engagement (20–57%) Generally gave positive feedback about their experiences and perceived usefulness	Not reported	Not reported
Atujuna et al., 2021 [28]	***Individual*** High engagement (100%); platform was “easy to use “; enjoyed communication “anytime, anywhere”; few network coverage issues. Participants liked that group facilitators responded to support needs.	Not reported	Not reported
Williams et al., 2023 [35]	Not reported	***Disease self-management*** CGM creates a sense of control over diabetes management; CGM assists in incorporating diabetes management into their identity.	***Individual health*** CGM created opportunities for positive outcomes.
Moffett et al., 2022 [32]	N/A (ongoing)	N/A (ongoing)	N/A (ongoing)
Mulawa et al., 2023 [29]	***Individual*** High engagement (83% used the app every day). Findings indicated that participants collectively spent 4.3 h in MASI, averaging 21.4 (range 1–50.8) minutes each.	Not reported	Not reported
Bantjes et al., 2024 [33]	***Individual*** Qualitative evidence indicated that there should be a keen focus on mental health and that the app reduced feelings of isolation. ***Health systems:*** The usability of the app increased through improving technical issues of the app by adding crosslinks within app activities.	***Promotion and prevention*** Risk behavior (suicidal ideation) was identified and SOPs were made to decrease the chance of risky behavior.	***Individual health*** Significantly higher adjusted risk differences (ARD; primary outcome) in joint anxiety/depression remission at 3 months and 6 months for Remote Group CST (ARD = 23.3–18.9%, *p* = 0.001–0.035) and SuperBetter (ARD = 12.7–22.2%, *p =* 0.047–0.006) than MoodFlow
Hacking et al., 2019 [30]	***Individual and health systems*** Participants found the virtual mentorship program engaging and appreciated having relatable, inspiring peers with whom they could openly communicate. Mentors were satisfied with the experience, reporting no issues with burden or difficulty in providing support to newly diagnosed youths.	***Disease self-management*** For the adolescents receiving virtual mentorship (VM), linkage to ART care was 28 (80%) vs. 30 (43%), and their retention in care was higher, 23 (92%) vs. 17 (89%), than the other cohort.	***Individual health*** VL suppression was slightly higher for the cohort receiving virtual mentorship: 26 (93%) vs. 29 (91%).

**Table 4 ijerph-22-00002-t004:** Contextual factors considered in the design of the digital health intervention.

Author and Year	Individual	Familial	Environmental
Zanoni et al., 2024 [31]	Age; language, health literacy	N/A	Safety, security, and privacy
Henwood et al., 2016 [27]	Age	N/A	Data/web access and cost; safety, security, and privacy
Atujuna et al., 2021 [28]	Age; health literacy	N/A	N/A
Williams et al., 2023 [35]	Age; language; device ownership/sharing/restrictions; technological ability	Beliefs about technology	Geographical location; data/web access and cost
Moffett et al., 2022 [32]	Age; language; technological ability	Socio-economic status	Geographical location; legal age, consent; safety, security and, privacy
Mulawa et al., 2023 [29]	Health literacy; language	N/A	Safety, security, and privacy
Bergam et al., 2022 [34]	Health literacy; age	Beliefs about technology	Data/web access and cost; legal age, consent
Bantjes et al., 2024 [33]	Technological ability; health literacy	N/A	N/A
Hacking et al., 2019 [30]	Age	N/A	Technology culture

## Data Availability

No new data were created or analyzed in this study. Data sharing is not applicable to this article.

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
