# Peer review of "Digital Health Interventions for Adolescents with Long-Term Health Conditions in South Africa: A Scoping Review"

_ijerph, 2024, doi:10.3390/ijerph22010002_

Round 1

Reviewer 1 Report

Comments and Suggestions for Authors

Dear Respectable Authors

Thank you for considering a great area of research related to Digital health. You investigated the digital health interventions (DHIs) for adolescents with long-term health conditions (LTHC) in South Africa to map the evidence in this field for better decision-making in your context. You employed a scoping review methodology to examine and present the breadth of evidence on this topic. Although your results are interesting, the way you report your manuscript needs some revisions as follows. I hope these recommendations will promote the quality of your manuscript. 

Cheers

- Please remove all subheadings from your abstract section.

- You stated that you searched the main databases but Wiley Online is not a database and it is a publisher. It is better to remove this or correct the way you stated it. You can exclude it from databases and add it as a supplementary search in publisher websites. 

- Please add more specific results to the abstract section. You have presented a lot of interesting results in the figures and tables, and you can add the most important parts of them in the abstract. Reduce the purely descriptive results and replace them with specific results, for example, on components, parameters, factors, etc.

- Your conclusion is not enough. Considering that you raised three questions at the end of the introduction, you should give a simple and clear answer to these questions in non-statistical language in the conclusion section.

- Please remove the subheadings of 1.1. and 1.2. from the introduction section. 

- Considering the e time frame that you stated in line 123, please correct lines 17-19 of the abstract and add 2014 to the period of search. 

- Why do you exclude the published article before 2014? Is there a rational reason? Please state here. In my opinion, DHIs-related interventions have been in use in most countries for more than ten years. With this limitation, you may have publication bias and miss some relevant articles unless you have a compelling reason to do so.

- You have not searched the two main databases, Scopus and the Web of Sciences. What is the reason? Failure to search these databases can also result in missing some relevant articles. Of course, reference mining and also Google Scholar could have reduced this bias, which is a good thing, but it would have been better if the main databases had been searched.

- The results obtained using this strategy you propose in PubMed do not match what is shown in Figure 1. Are you sure the numbers are correct? With this search strategy, you have retrieved 3250 records in PubMed. I employed your strategy and the results of the search are only about 600 records. 

- Other search strategies are not mentioned and the corresponding numbers in Figure 1 do not make much sense. How did you search in Scholar and only 14 records were retrieved? What do you mean by unspecified in Figure 1? Please add search strategies for all databases according to PRISMA-ScR with the number of records and the exact time of the search. It is better to add it as a supplementary file. 

- Line 149, please remove Figure 1 from the methods section and add it in the results section under the "Selection of sources of evidence" subheading (item 14 of PRISMA-ScR). 

- The order of the Figures in the text is not followed. Figure 2 is referred to first and then Figure 1. Figure 2 should be removed from this section. There is also no need to include Figure 1 in the text of this article. You can upload it as a supplementary file.

- You stated that follow PRISMa-ScR but some items did not match this checklist. For example, in the scoping review, we have a data charting process, not data extraction. Please correct it considering items 10 and 11 of the PRISMA-ScR checklist. 

- Please remove lines 166-168. Since this item is not required in scoping reviews, there is no need to include a reason for not doing it.

- The sum of the numbers shown in the diagram (3574) is not the same as the number written in the text (3570). Please check it again. You screened 3574 records, not 3570. 

- Subheading 3.1. needs to be modified. Considering PRISMA-ScR, you must write only Characteristics of sources of evidence under this subheading. Please separate the other results related to challenges in a separate subheading. 

- Line 190, Scheme 31.? This word is not clear to me. What do you mean by Scheme 31?

- Discussion section, please add your limitations, strengths, and weaknesses of your study. Also, prepare some practical recommendations based on your results and also for future research. 

- It is best to discuss each of your study questions in three separate sections in the discussion section of your paper. Your discussion is currently not very strong and needs to be improved. You should discuss two types: an internal discussion that compares the included articles in this study, and an external discussion that compares your results with similar studies outside your setting.

- In conclusion, you should first provide a direct response to your study questions in a way that is understandable to a general audience, and in addition, make recommendations based on your results.

Author Response

Reviewer 1

Dear Respectable Authors

Thank you for considering a great area of research related to Digital health. You investigated the digital health interventions (DHIs) for adolescents with long-term health conditions (LTHC) in South Africa to map the evidence in this field for better decision-making in your context. You employed a scoping review methodology to examine and present the breadth of evidence on this topic. Although your results are interesting, the way you report your manuscript needs some revisions as follows. I hope these recommendations will promote the quality of your manuscript. 

Cheers

Comment

- Please remove all subheadings from your abstract section.

Response

Thank you. The headings were removed.

Comment

- You stated that you searched the main databases but Wiley Online is not a database and it is a publisher. It is better to remove this or correct the way you stated it. You can exclude it from databases and add it as a supplementary search in publisher websites. 

Response

Wiley Online Library was used, which was corrected in the abstract and in line 145. It serves as a database to access articles and publications.

Comment

- Please add more specific results to the abstract section. You have presented a lot of interesting results in the figures and tables, and you can add the most important parts of them in the abstract. Reduce the purely descriptive results and replace them with specific results, for example, on components, parameters, factors, etc.

Response

Thank you. The abstract was revised accordingly.

Comment

- Your conclusion is not enough. Considering that you raised three questions at the end of the introduction, you should give a simple and clear answer to these questions in non-statistical language in the conclusion section.

Response

Thank you. The abstract was revised.

Comment

- Please remove the subheadings of 1.1. and 1.2. from the introduction section. 

Response

The subheadings were removed.

Comment

- Considering the e time frame that you stated in line 123, please correct lines 17-19 of the abstract and add 2014 to the period of search. 

Response

The change was made in line 20 of the abstract to include the 2014.

Comment

Why do you exclude the published article before 2014? Is there a rational reason? Please state here. In my opinion, DHIs-related interventions have been in use in most countries for more than ten years. With this limitation, you may have publication bias and miss some relevant articles unless you have a compelling reason to do so.

Response

This review focused on the current evidence of digital interventions for long-term health conditions, therefore in order to yield relevant results we did not include studies published prior to 2014. We added a rationale for the timeframe – lines 127-129.

Comment

- You have not searched the two main databases, Scopus and the Web of Sciences. What is the reason? Failure to search these databases can also result in missing some relevant articles. Of course, reference mining and also Google Scholar could have reduced this bias, which is a good thing, but it would have been better if the main databases had been searched.

Response

The JBI Manual for Scoping Reviews recommends the use of MEDLINE and CINAHL. The JBL Guidelines further recommend a minimum usage of three multidisciplinary and topic-specific databases, and this recommendation was therefore met with the chosen databases. Changes made in lines 143-145.

Comment

- The results obtained using this strategy you propose in PubMed do not match what is shown in Figure 1. Are you sure the numbers are correct? With this search strategy, you have retrieved 3250 records in PubMed. I employed your strategy and the results of the search are only about 600 records. 

Response

The search was run again in PubMed after recommendation from the reviewer, and 5463 results were yielded. We believe that the search was done correctly and verified by an information specialist. The discrepancy may be due to database updates, search settings, or variations in how the strategy was applied. The larger yield aligns with the exploratory nature of scoping reviews, prioritising sensitivity.

Comment

- Other search strategies are not mentioned and the corresponding numbers in Figure 1 do not make much sense. How did you search in Scholar and only 14 records were retrieved? What do you mean by unspecified in Figure 1? Please add search strategies for all databases according to PRISMA-ScR with the number of records and the exact time of the search. It is better to add it as a supplementary file. 

Response

We used the same PubMed search string for Google Scholar, and relevant studies were selected by hand by the reviewers. Amended on line 150.

Searches were run on 29 July 2024. Added in line 184.

“Unspecified” was fixed in the PRISMA Diagram (Figure 2)

We have added the database results in lines 187-189.

Comment

- Line 149, please remove Figure 1 from the methods section and add it in the results section under the "Selection of sources of evidence" subheading (item 14 of PRISMA-ScR). 

Response

The reference to Figure 2 was removed from the methods. The sub-heading “Selection of sources of evidence” added to the results section. Line 183.

Comment

- The order of the Figures in the text is not followed. Figure 2 is referred to first and then Figure 1. Figure 2 should be removed from this section. There is also no need to include Figure 1 in the text of this article. You can upload it as a supplementary file.

Response

Reference to Figure 2 was removed from the methods. Figure 2 is maintained in the article.

Comment

- You stated that follow PRISMa-ScR but some items did not match this checklist. For example, in the scoping review, we have a data charting process, not data extraction. Please correct it considering items 10 and 11 of the PRISMA-ScR checklist. 

Response

Data extraction changed to data charting. Line 159. The 2024 JBI indicate that: In scoping reviews, the data extraction process may be referred to as “data charting”.

Comment

- Please remove lines 166-168. Since this item is not required in scoping reviews, there is no need to include a reason for not doing it.

Response

Thank you, the lines were removed.

Comment

- The sum of the numbers shown in the diagram (3574) is not the same as the number written in the text (3570). Please check it again. You screened 3574 records, not 3570. 

Response

As evidenced in the PRISMA diagram (Figure 2), 4 studies were automatically removed by Covidence as they were duplicates. These studies were removed before screening occurred. The clarification has been added in lines 184-185.

Comment

- Subheading 3.1. needs to be modified. Considering PRISMA-ScR, you must write only Characteristics of sources of evidence under this subheading. Please separate the other results related to challenges in a separate subheading. 

Response

The sub-heading was modified. Line 194. We decided to keep the column and information related to challenges as it is the challenge addressed by the intervention which further describes the focus of the intervention and thus the study characteristics. We believe that we still comply with the PRISMA-ScR guidelines as the guidelines indicate the minimum information to be extracted.

Comment

- Line 190, Scheme 31.? This word is not clear to me. What do you mean by Scheme 31?

Response

We are not sure what the reviewer is referring to. We searched the document and could not find ‘Scheme 31’.

Comment

- Discussion section, please add your limitations, strengths, and weaknesses of your study. Also, prepare some practical recommendations based on your results and also for future research. 

Response

We have added information about the strengths, limitations and practical recommendations. We recommend that future research should include studies that implement and evaluate DHI’s for adolescents with long-term health conditions. Additionally, there is a gap in developing DHIs that are tailored to South Africa’s diverse population. Designing interventions with adolescents in South Africa along with stakeholders like the community clinics could ensure the digital tools meet the needs of adolescents.

Comment

- It is best to discuss each of your study questions in three separate sections in the discussion section of your paper. Your discussion is currently not very strong and needs to be improved. You should discuss two types: an internal discussion that compares the included articles in this study, and an external discussion that compares your results with similar studies outside your setting.

Response

Thank you. The discussion of this review study now has been separated into three sections. The first one addresses the first research question which focuses on the Digital Health Interventions that have been studied for adolescents with long-term health conditions in South Africa. The external studies which compare to the findings of the study were highlighted. However, there is a gap in the internal studies that support the study findings. This was noted as a limitation.  The second section focuses on the long-term health conditions targeted by the DHIs. While the third one focuses on the reported outcomes of the DHIs. See lines 341-443.

Comment

- In conclusion, you should first provide a direct response to your study questions in a way that is understandable to a general audience, and in addition, make recommendations based on your results.

Response

Thank you. We have responded to the research questions in the discussion section and included recommendations based on the results.

Reviewer 2 Report

Comments and Suggestions for Authors

Page 2, line 51/52:

‘In South Africa, HIV and other non-communicable diseases (NCDs) account for 38% of deaths among adolescent girls aged 15 to 19 years and 29% of deaths among adolescent boys in the same age group [5].’

Do you mean HIV and other communicable diseases, as HIV is communicable?

Figure 1:

You’ve made a good case for focusing specifically on South Africa (rather than, say, southern Africa more widely given migration etc.) Your logic model, however, is very Minority World-focused both in the sense of the theoretical underpinnings, and in relation to the outcome measures. For example, how does cultural or spiritual orientation, or even traditional medicine, play into the factors you were examining in the review? And if they don’t, should they?

Discussion:

Again coming to the SA-specificity of this review, it was quite difficult to see any aspects of the discussion that could not have been applied to any southern African context, or even specific to low-resource contexts. For example, how do users of these e-health interventions manage access to data and charge for devices? I appreciate of course that if the articles you analysed did not speak to the context in any depth then that makes the task more difficult for you, but perhaps you could use the Discussion to challenge this significant gap in the literature?

Similarly, your analysis does not share any findings relating to gender, urban/rural, or other important contextual factors. I think digging into these factors and identifying if there are any particular gaps, homogeneity vs heterogeneity of the studies’ populations would deepen your analysis.

Author Response

Comment

Page 2, line 51/52:

‘In South Africa, HIV and other non-communicable diseases (NCDs) account for 38% of deaths among adolescent girls aged 15 to 19 years and 29% of deaths among adolescent boys in the same age group [5].’

Do you mean HIV and other communicable diseases, as HIV is communicable?

Response

Thank you. “Other” was removed from the sentence.

Comment

Figure 1:

You’ve made a good case for focusing specifically on South Africa (rather than, say, southern Africa more widely given migration etc.) Your logic model, however, is very Minority World-focused both in the sense of the theoretical underpinnings, and in relation to the outcome measures. For example, how does cultural or spiritual orientation, or even traditional medicine, play into the factors you were examining in the review? And if they don’t, should they?

 Response

The logic model specifically focuses on contextual factors that influence the implementation of digital health interventions. As this was a scoping review, the analysis aimed to map key components and contextual factors in line with the objectives of scoping reviews, rather than to conduct an in-depth analysis of all possible influences. While cultural, spiritual, and traditional medicine factors were not explicitly addressed in the logic model, we acknowledge their potential significance in the South African context and note this as a limitation of the current approach. Lines 433-434.

Comment

Discussion:

Again coming to the SA-specificity of this review, it was quite difficult to see any aspects of the discussion that could not have been applied to any southern African context, or even specific to low-resource contexts. For example, how do users of these e-health interventions manage access to data and charge for devices? I appreciate of course that if the articles you analysed did not speak to the context in any depth then that makes the task more difficult for you, but perhaps you could use the Discussion to challenge this significant gap in the literature?

Response

Thank you. In the first section of the discussion, the researchers added information addressing the South African and SSA context, particularly on the aspect of data use and cost in this context. The significant gap in the adoption of the DHIs as a result of the high data costs was also highlighted.  Lines 363-365.

Comment

Similarly, your analysis does not share any findings relating to gender, urban/rural, or other important contextual factors. I think digging into these factors and identifying if there are any particular gaps, homogeneity vs heterogeneity of the studies’ populations would deepen your analysis.

Response

Thank you. Consideration of geographical context in the development of the intervention was mentioned in Table 4. We added information about rural/urban implementation to Table 1 as well as the gender distribution where reported. While our analysis recognises the potential importance of these contextual factors, the primary focus of our logic model was on key factors identified as directly influencing the implementation of digital health interventions for adolescents. Gender, in particular, was not highlighted as a significant factor in the studies reviewed, but we acknowledge that this represents a gap in the literature that warrants further exploration.

We agree that a deeper analysis of homogeneity versus heterogeneity in study populations, including gender and urban/rural distinctions, could enhance future work in this area. However, the research in this area is still limited.

Reviewer 3 Report

Comments and Suggestions for Authors

This study aimed to describe digital health interventions among African adolescents with chronic diseases

Materials and Methods

Concerning inclusion criteria, authors must specify why studies on adolescents aged 15-24 were included, since they are studies and not participants

Regarding Figure 2, authors must explain the reason why 3550 studies were ruled out, since there is no explanation.

Author Response

Reviewer 3

This study aimed to describe digital health interventions among African adolescents with chronic diseases

Comment

Materials and Methods

Concerning inclusion criteria, authors must specify why studies on adolescents aged 15-24 were included since they are studies and not participants

Response

The JBI Scoping review guidelines were followed which indicate that the type of participants must be identified. The guidelines state: Important characteristics of participants should be detailed, including age and other qualifying criteria that make them appropriate for the objectives of the scoping review and for the review question.

Comment

Regarding Figure 2, the authors must explain the reason why 3550 studies were ruled out, since there is no explanation.

Response

Thank you for the comment. We have clarified this on the PRISMA figure. (Figure 2) Studies were excluded during the title and abstract screening due to irrelevance.

Round 2

Reviewer 1 Report

Comments and Suggestions for Authors

Dear Respectable Authors

Thank you for your clarifications.

Cheers

Reviewer 2 Report

Comments and Suggestions for Authors

Thank you for your thoughtful responses!